# Safety and Immunogenicity of Co-Administration of Herpes Zoster Vaccines with Other Vaccines in Adults: A Systematic Review and Meta-Analysis

**DOI:** 10.3390/vaccines13060637

**Published:** 2025-06-12

**Authors:** Omid Rezahosseini, Aysan Bazargan, Mads Frederik Eiberg, Alexander Printzlau Korsgaard, Raziyeh Niyati, Christina Ekenberg, Lars Nørregaard Nielsen, Zitta Barrella Harboe

**Affiliations:** 1Department of Pulmonary and Infectious Diseases, Copenhagen University Hospital North Zealand, Dyrehavevej 29, 3400 Hillerød, Denmarkmads.frederik.eiberg.01@regionh.dk (M.F.E.); alexander@korsgaard.org (A.P.K.); raziyeh.niyati@regionh.dk (R.N.); christina.ekenberg.01@regionh.dk (C.E.); lars.noerregaard.nielsen@regionh.dk (L.N.N.); 2Department of Clinical Medicine, Faculty of Health and Medical Science, University of Copenhagen, Blegdamsvej 3B, 2200 Copenhagen, Denmark; 3Department of Bacteria, Parasites, and Fungi, Statens Serum Institute, Artillerivej 5, 2300 Copenhagen, Denmark

**Keywords:** herpes zoster, vaccination, co-administration, safety, immune response, systematic review

## Abstract

**Introduction:** Herpes zoster (HZ), or shingles, is a vaccine-preventable disease with two approved vaccines: the live-attenuated vaccine (LZV) and the adjuvanted recombinant zoster vaccine (RZV). Evidence on the immunogenicity and adverse events (AEs) following co-administration with other vaccines in adults is limited. This systematic review and meta-analysis aims to evaluate the immunogenicity and safety of HZ vaccines when co-administered with other vaccines. **Methods:** We followed PRISMA 2020 guidelines and systematically searched multiple databases (January 1950 to February 2024) for studies on HZ vaccination with concomitant vaccines in adults (≥18 years). Observational studies, randomized controlled trials (RCTs), and non-randomized controlled trials were included, excluding reviews, case series, case reports, editorials, and non-English publications. Risk of bias was assessed using Cochrane tools (RoB 2 and ROBINS-I). A meta-analysis compared geometric mean concentration (GMC) ratios and vaccine response rates (VRRs) for RZV, applying the Hartung–Knapp adjustment. For LZV, meta-analysis was not feasible, and results were described narratively. AEs were analyzed using risk ratios and presented in forest plots. **Results:** Out of 369 search hits, ten RCTs were included. In six RCTs, RZV was co-administered with influenza, COVID-19, pneumococcal vaccines (PCV13, PPSV23), or Tdap. The pooled GMC mean difference was −0.04 (95% CI: −0.10 to 0.02, *p* = 0.19), and the pooled VRR was 1.00 (95% CI: 0.99 to 1.01, *p* = 0.59). Local and systemic AEs showed pooled relative risks of 0.99 (95% CI: 0.95 to 1.03, *p* = 0.73) and 1.01 (95% CI: 0.91 to 1.11, *p* = 0.90), respectively. LZV co-administration was investigated in four RCTs and was safe; however, co-administration with PPSV23 resulted in reduced immunogenicity. **Conclusions:** The co-administration of RZV with other vaccines was safe and immunogenic. However, limited evidence suggests that co-administration of LZV with PPSV23 reduced the immunogenicity of LZV through an unknown mechanism. Still, RZV co-administration could enhance vaccine uptake in vulnerable populations.

## 1. Introduction

The varicella-zoster virus (VZV) infection typically results in acute varicella, commonly known as chickenpox. Following the primary infection, the virus can remain latent in the dorsal root ganglia and reactivate later in approximately one-third of the population during their lifetime. This reactivation is called herpes zoster (HZ), and the incidence of HZ increases with advancing age or in individuals with compromised immune systems [1]. HZ can lead to complications such as post-herpetic neuralgia, and in vulnerable individuals, it can progress to severe, disseminated infection, resulting in hospitalization or death. Therefore, preventing HZ is important, and international guidelines recommend HZ vaccination for immunocompromised adults aged >18 years and all adults aged ≥50 years [2]. Currently, two HZ vaccine platforms have been approved: the live-attenuated VZV vaccine (LZV, Zotavax^®^, MSD) and the adjuvanted VZV glycoprotein E (gE) subunit vaccine (rVZV, Shingrix, GSK) [3,4,5]. Zostavax^®^ is indicated for immunocompetent adults aged ≥50 years, but due to its live attenuated nature, it is contraindicated for immunocompromised individuals. In contrast, Shingrix, a non-live recombinant vaccine, is recommended for adults aged ≥50 years and immunocompromised adults aged ≥18 years due to its favorable safety profile and immunogenicity in these populations [3,4,5].

Due to the overlap in targeted age groups, immune status, and clinical recommendations, individuals who receive HZ vaccines are also usually candidates to receive other vaccines, such as seasonal influenza, pneumococcal, and COVID-19 [6]. Co-administration not only simplifies vaccination schedules, and enhances patient compliance and convenience, but also improves vaccine uptake and coverage rates, which are essential for effective prevention of infectious diseases in vulnerable populations [7]. However, evidence regarding the immunological responses and adverse events following the co-administration of HZ vaccines with other vaccines in adults is limited. Furthermore, most of our general understanding and evidence about vaccine co-administrations are derived from studies conducted in children [7]. However, the immune response to vaccines can vary in adults due to comorbidities, immunosuppression, or immunosenescence [8]. Hence, further investigation and data collection on the co-administration of HZ vaccines with other vaccines is required.

We systematically reviewed the existing evidence on the safety and immunogenicity of HZ vaccines when they are co-administered with other vaccines in adults. The findings can support evidence-based recommendations for co-administration practices and improve our understanding of the co-administration of these vaccines in at-risk adults.

## 2. Materials and Methods

### 2.1. Search Strategy

This systematic review was conducted following the guidelines outlined by the Preferred Reporting Items for Systematic Reviews and Metaanalyses (PRISMA 2020) statement. The primary clinical question explored was as follows: “Is there a difference in the safety and immunogenicity when HZ vaccines are co-administered with other vaccines compared to when administered separately?”.

To formulate this question, we followed the Population, Intervention, Comparator, Outcome, and Study design (PICOS) process. An extensive literature search was carried out in several databases, including PubMed, MEDLINE, EMBASE, Scopus, and Web of Science, covering the period from 1 January 1950 to 12 December 2024. Two independent reviewers (AB and OR) performed the initial screening of papers using predetermined search terms, evaluating titles and abstracts for relevance. Relevant papers were then thoroughly assessed in full text, applying inclusion criteria. Any disagreements arising during screening or inclusion were settled by consultation with a third independent reviewer (ZBH), ensuring consistent and objective decisions. Additionally, reference lists of the included articles were examined, and supplementary manual searches were performed using Google to uncover further relevant studies potentially missed by the electronic searches.

The study protocol for this systematic review was registered on the Open Science Framework (OSF) (https://osf.io/z9yr5 (accessed on 6 August 2024)). OSF is a free, open-source platform designed to support the research lifecycle, facilitating open collaboration in scientific research by providing a platform for conducting, managing, and sharing work more transparently. By following these systematic procedures, we aimed to maintain the integrity and transparency of our research process.

### 2.2. Inclusion Criteria

HZ vaccines were defined as both the LZV and RZVs, and we included studies on HZ vaccination where vaccines for any other infection were concomitantly administered to adults (≥18 years). We considered observational studies, randomized controlled trials, and non-randomized controlled trials, excluding reviews, case series, case reports, ideas, editorials, and opinions. Moreover, we excluded studies in non-human subjects and those published in languages other than English.

### 2.3. Full Search Strategy

We employed a combination of MeSH terms and conducted a separate search for free-text terms in the PubMed/MEDLINE databases. The following MeSH term combination was used in PubMed, resulting in 21 hits: (“Herpes Zoster Vaccine”[Mesh]) AND (“Influenza Vaccines”[Mesh] OR “COVID-19 Vaccines”[Mesh] OR “Papillomavirus Vaccines”[Mesh] OR “Meningococcal Vaccines”[Mesh] OR “Pneumococcal Vaccines”[Mesh] OR “Diphtheria-Tetanus-acellular Pertussis Vaccines”[Mesh] OR “Hepatitis B Vaccines”[Mesh] OR “Tuberculosis Vaccines”[Mesh] OR “Hepatitis A Vaccines”[Mesh] OR “Haemophilus Vaccines”[Mesh] OR “Streptococcal Vaccines”[Mesh] OR “Mumps Vaccine”[Mesh] OR “Measles-Mumps-Rubella Vaccine”[Mesh] OR “Diphtheria-Tetanus Vaccine”[Mesh] OR “Yellow Fever Vaccine”[Mesh] OR “Diphtheria-Tetanus-Pertussis Vaccine”[Mesh] OR “Viral Hepatitis Vaccines”[Mesh] OR “Pertussis Vaccine”[Mesh] OR “Measles Vaccine”[Mesh] OR “BCG Vaccine”[Mesh] OR “BNT162 Vaccine”[Mesh] OR “2019-nCoV Vaccine mRNA-1273”[Mesh] OR “Heptavalent Pneumococcal Conjugate Vaccine”[Mesh] OR “Human Papillomavirus Recombinant Vaccine Quadrivalent, Types 6, 11, 16, 18”[Mesh] OR “ChAdOx1 nCoV-19”[Mesh] OR “SARS-CoV-2 inactivated vaccines” [Supplementary Concept] OR “diphtheria-tetanus-acellular pertussis-Hib-hepatitis B vaccine” [Supplementary Concept]).

For the free-text search, we used the following combination of search terms, resulting in 23 hits in PubMed: (“Herpes Zoster Vaccine” OR “Shingrix” OR “Zostavax”) AND (“Influenza Vaccines” OR “COVID-19 Vaccines” OR “Papillomavirus Vaccines” OR “Meningococcal Vaccines” OR “Pneumococcal Vaccines” OR “Diphtheria-Tetanus-acellular Pertussis Vaccines” OR “Hepatitis B Vaccines” OR “Tuberculosis Vaccines” OR “Hepatitis A Vaccines” OR “Haemophilus Vaccines” OR “Streptococcal Vaccines” OR “Mumps Vaccine” OR “Measles-Mumps-Rubella Vaccine” OR “Diphtheria-Tetanus Vaccine” OR “Yellow Fever Vaccine” OR “Diphtheria-Tetanus-Pertussis Vaccine” OR “Viral Hepatitis Vaccines” OR “Pertussis Vaccine” OR “Measles Vaccine” OR “BCG Vaccine” OR “BNT162 Vaccine” OR “2019-nCoV Vaccine mRNA-1273” OR “Heptavalent Pneumococcal Conjugate Vaccine” OR “Human Papillomavirus Recombinant Vaccine Quadrivalent, Types 6, 11, 16, 18” OR “ChAdOx1 nCoV-19” OR “SARS-CoV-2 inactivated vaccines” OR “diphtheria-tetanus-acellular pertussis-Hib-hepatitis B vaccine”).

All search results were imported into Covidence (https://www.Covidence.Org/, accessed on 8 April 2025). (Covidence. Covidence—Better Systematic Review Management. 2024. Available at: https://www.Covidence.Org/, accessed on 8 April 2025). Covidence is a web-based tool that facilitates the systematic review process.

### 2.4. Data Extraction and Risk of Bias

We extracted data using Extraction 2.0 on the Covidence platform. Investigators were invited via email to participate in the project on the platform, which helps remove duplicate studies found across several databases. Each investigator independently screened the imported studies by title and abstract, including those deemed relevant for full-text review or excluding them with a reason. Data extraction forms were designed on the platform, and data were extracted directly within it.

We assessed the risk of bias using the Cochrane Risk of Bias 2 for randomized clinical trials (RoB 2) and the Risk of Bias in Non-randomized Studies of Interventions (ROBINS-I). The risk-of-bias visualization tool (Robvis) was used to visualize the quality of the included studies with traffic light plots. These tools utilize standard signaling questions to identify potential bias within a study systematically.

The risk of bias was categorized as low, some concerns, or high for randomized clinical trials and low, moderate, or high for non-randomized clinical trials. Due to the heterogeneity of the included studies, a meta-analysis could not be performed, and the data were summarized narratively.

### 2.5. Meta-Analysis

We conducted separate meta-analyses for immunogenicity and adverse effects. We used geometric mean concentration (GMC), or vaccine response rate (VRR) values to investigate immunogenicity. The proportion of AEs was used to investigate local and systemic AEs. Studies that reported values for both co-administration and single-administration groups were included in the meta-analysis. The common-effect model (CEM) and random-effects model (REM) were utilized, with the Hartung–Knapp adjustment applied to provide more accurate confidence intervals. Moreover, forest plots were generated to display the estimates.

Heterogeneity across studies was evaluated using statistical and visual methods. The I^2^ statistic was used to quantify the proportion of variability attributable to heterogeneity rather than chance. The Tau^2^ statistic was used to estimate between-study variance within the random-effects model. Funnel plots were used to detect potential publication bias. Heterogeneity was classified as follows: 0% to 40% as “might not be important”; 30% to 60% as “may represent moderate heterogeneity”; 50% to 90% as “may represent substantial heterogeneity”; and 75% to 100% as “considerable heterogeneity” [9]. All analyses were performed using R version 4.1.0 with the meta package.

## 3. Results

We reviewed 369 articles and included ten RCTs that fulfilled the inclusion criteria in our study (Figure 1). Six RCTs investigated RZV co-administered with influenza vaccines, the COVID-19 mRNA vaccine, pneumococcal vaccines (PCV13, PPSV23), or Tdap. Four RCTs investigated LZV, which was co-administered with influenza vaccines or PPSV23. The characteristics of the studies are summarized in Table 1.

### 3.1. Live Zoster Vaccines: Immunogenicity and Adverse Effects

Due to the low number and heterogeneity of studies, it was not accurate to perform a meta-analysis on immunogenicity data of LZVs. Levin et al. included 882 participants and compared the concomitant and sequential administration of LZV and inactivated influenza vaccine (IIV4) in adults 50 and older. Participants were randomized into two groups: the concomitant group, which received LZV and IIV4 at separate injection sites on day 1 and placebo at week 4, and the sequential group, which received IIV4 and a placebo on day 1 and LZV at week 4 [15]. The VZV antibody GMT ratio (concomitant/sequential) was 0.87 (95% CI: 0.80–0.95), meeting the prespecified non-inferiority criterion, while the VZV geometric mean fold-rise (GMFR) in concomitant was 1.9 (95% CI: 1.76–2.05), meeting the acceptability criterion [15].

Kerzner et al. randomized participants to receive LZV concomitantly with influenza vaccine on day 1 at separate injection sites, followed by placebo at week 4, or sequentially with a placebo on day 1 and LZV at week 4. The GMT ratio (0.9, 95% CI: 0.8–1.0) demonstrated non-inferiority, with an acceptable GMFR of 2.1 (95% CI: 2.0–2.3) in the concomitant group [17].

In the study by MacIntyre et al., 473 participants aged 60 years or older were randomized to receive LZV and PPV23 either concomitantly on day 1 or separately, with PPV23 on day 1 and LZV at week 4. VZV antibody response was lower in the concomitant group than in the non-concomitant group, with an estimated GMT ratio of 0.70 (95% CI: 0.61–0.80) [18].

Hata et al. included 54 individuals with diabetes and randomized them to receive a high-dose LZV with PPSV23 or a placebo with PPSV23 [3]. It should be noted that the live, attenuated Oka varicella vaccine^®^ was used in this study, which was different from ZOSTAVAX^®^. Cell-mediated and humoral immunity were investigated using the VZV skin test and immune adherence haemagglutination (IAHA) titers, respectively. Furthermore, the secondary analyses included an interferon-γ ELISPOT assay. The changes in skin test scores were not significantly different (*p* = 0.2155). Moreover, the GMT did not increase three months after vaccination in either group. The GMFR in ELISPOT counts were identical at 1.2 in both groups, with overlapping confidence intervals and no significant difference (*p* = 0.989) [3].

In all four RCTs, the frequency of vaccine-related AEs, including injection-site and systemic reactions, was similar between concomitant and sequential groups, and no significant vaccine-related serious AEs were observed [3,8,15,17].

### 3.2. Recombinant Zoster Vaccines: Immunogenicity and Adverse Effects

We performed the meta-analysis on five studies examining the GMC ratios and VRR of the RZVs. The pooled GMC mean difference was −0.04 (95% CI: −0.10 to 0.02, *p* = 0.19, I^2^ = 0.0%), and the pooled VRR was 1.00 (95% CI: 0.99 to 1.01, *p* = 0.59, I^2^ = 0.0%) (Figure 2).

Local and systemic AEs were investigated in six RCTs; however, RCTs by Schwarz et al. [14] and Maréchal et al. [14] were excluded from the meta-analysis due to heterogenicity, and the study by Schmader et al. [10] was not included due to the absence of a single HZ vaccination control group. The meta-analysis of three RCTs showed pooled relative risks of 0.99 (95% CI: 0.95 to 1.03, *p* = 0.73, I^2^ = 35%) and 1.01 (95% CI: 0.91 to 1.11, *p* = 0.90, I^2^ = 24%), respectively (Figure 3).

### 3.3. Risk of Bias

The risk of bias was low in nine of the ten included RCTs, while the only study with some concerns had issues arising from missing outcome data.

## 4. Discussion

We systematically reviewed studies regarding the immunogenicity and safety of zoster vaccines co-administered with other vaccines. Studies generally had a low risk of bias, and the vaccines were co-administered safely. Our findings highlighted the distinct immunogenicity potentials of LZVs and RZVs in combination with other vaccines. LZVs showed a non-inferior variability in immune responses; however, the pooled analysis of RZVs showed consistent immunogenicity.

For LZVs, our results are more variable, reflecting heterogeneity in study designs and outcomes. Two RCTs reported non-inferiority in immune responses when LZV was co-administered with influenza vaccines compared to single administration [8,15]. However, the RCT by MacIntyre et al. in 2010 raised concerns about reduced immunogenicity following the co-administration of LZV and PPSV23 [18]. The LZV (Zostavax^®^) contains a mean titer of 19,000 plaque-forming units per dose (PFU). In 2015, Hata et al. utilized a high-dose LZV containing an estimated 50,000 plaque-forming units per dose derived from the Oka strain [3]. The immunogenicity was compared to the placebo; however, PPSV23 was co-administered with both LZV and the placebo, and the patients had diabetes [3]. The study by Hata et al. is the only study that investigated cellular immunity using interferon-γ ELISPOT assay [3], and cellular immunity is important in controlling VZV replication and preventing HZ [19]. Hata et al. concluded that co-administration of this high-dose LZV and PPSV23 can not improve immunity to VZV in older adults with diabetes [3]. Although this finding suggested a potential negative signal for the co-administration of LZV and PPSV23, its clinical importance remained unclear. This uncertainty was addressed in 2018 by a cohort study conducted by Bruxvoort et al., which included more than 35,000 individuals and found no significant difference in the risk of infection regardless of whether the LZV and PPSV23 vaccines were administered concomitantly or separately [20].

We found and included RCTs that investigated the co-administration of RZV with influenza vaccines, COVID-19 mRNA vaccines, pneumococcal vaccines (PCV13, PPSV23), and Tdap. Our meta-analysis supports the growing evidence that RZVs can be safely co-administered with other vaccines without compromising immunogenicity. Our findings are further supported by a recent comprehensive systematic review and meta-analysis by Losa et al., which specifically assessed the immunogenicity of RZV across various adult populations, including immunocompromised individuals [21]. Notably, their analysis of within-study GMC comparisons found non-inferiority of humoral responses following RZV co-administration with other vaccines (COVID-19 mRNA-1273, PCV13, PPSV23, Tdap, and IIV4). Moreover, in line with our findings, Losa et al. observed no significant differences in VRRs related to co-administration [21]. In another review, Omar Ali et al. concluded that RZV co-administration does not significantly affect immunogenicity [9]. Although the aims, clinical questions, and methods used in the mentioned studies [9,19] differed from ours, their findings further support that co-administered vaccines do not attenuate immune response to the RZV.

The absence of significant increases in local and systemic adverse events in our study is reassuring and supports using RZVs in multi-vaccine strategies. RZV is an adjuvanted vaccine that incorporates the AS01 adjuvant system, which enhances innate immune activation and induces robust CD4^+^ T-cell and humoral responses [22]. Other vaccines, such as the adjuvanted influenza vaccine Fluad^®^ Quadrivalent, adjuvanted hepatitis B vaccines (e.g., Heplisav-B^®^), and the RSV vaccine Arexvy^®^, also contain adjuvants. The co-administration of two adjuvanted vaccines may theoretically increase adverse events due to enhanced stimulation of the innate immune system. These effects may have clinical implications and could influence vaccine acceptance. Although such concerns are biologically plausible, current evidence to support this hypothesis is limited. A recent report from GSK presented preliminary results from the clinical trial NCT05966090, indicating acceptable reactogenicity following the co-administration of RZV and an RSV vaccine, though detailed data are not yet available.

The study by Schmader et al. was the only one that utilized RZV and adjuvanted influenza vaccine (aIIV4) and compared it to RZV and high-dose inactivated influenza vaccine (HD-IIV4) co-administration [10]. The RCT investigated safety but not immunogenicity and showed that these two vaccine combinations are safe and similar regarding the proportion of adverse effects [10]. Further studies are warranted to investigate the safety and immunologic interactions of co-administering two adjuvanted vaccines, especially in vulnerable populations.

A systematic approach, performing a meta-analysis, using a broad search term combination, and searching in different databases were among the strengths of our study. However, the limited number of LZV studies and their heterogeneity restricted our ability to draw firm conclusions about this vaccine type. Additionally, the included RCTs primarily involved healthy adults, potentially limiting the generalizability of these findings to populations with underlying health conditions or immunocompromised individuals.

## 5. Future Perspectives

Future studies can explore the immunogenicity and safety of additional vaccine combinations, such as RZVs with newer pneumococcal conjugate vaccines (e.g., PCV20), respiratory syncytial virus (RSV) mRNA vaccines, two adjuvanted vaccines, or live vaccines. It is also important to investigate the long-term immunological impact of co-administration, including the durability of immune responses and vaccine efficacy and effectiveness. Cellular immunity may impact the durability of vaccine responses and, therefore, needs to be investigated. Still, our findings align with current recommendations emphasizing the practicality of co-administration to enhance vaccine coverage in adult populations, particularly among older adults at increased risk for multiple vaccine-preventable diseases [7].

## 6. Conclusions

Our findings support the co-administration of RZVs with influenza vaccines, COVID-19 mRNA vaccines, pneumococcal vaccines (PCV13, PPSV23), and Tdap, showing no significant impact on immunogenicity or adverse events in healthy adults. While co-administration of LZVs was generally safe, the limited number of studies, and exclusion of immunocompromised populations highlight the need for further investigation to confirm safety and immunogenicity in the vulnerable populations. Our findings are consistent with previous research and offer practical implications for public health policies, supporting flexible and efficient vaccination schedules that can improve coverage, reduce missed opportunities for immunization, and enhance protection in aging and vulnerable populations.

## Figures and Tables

**Figure 1 vaccines-13-00637-f001:**
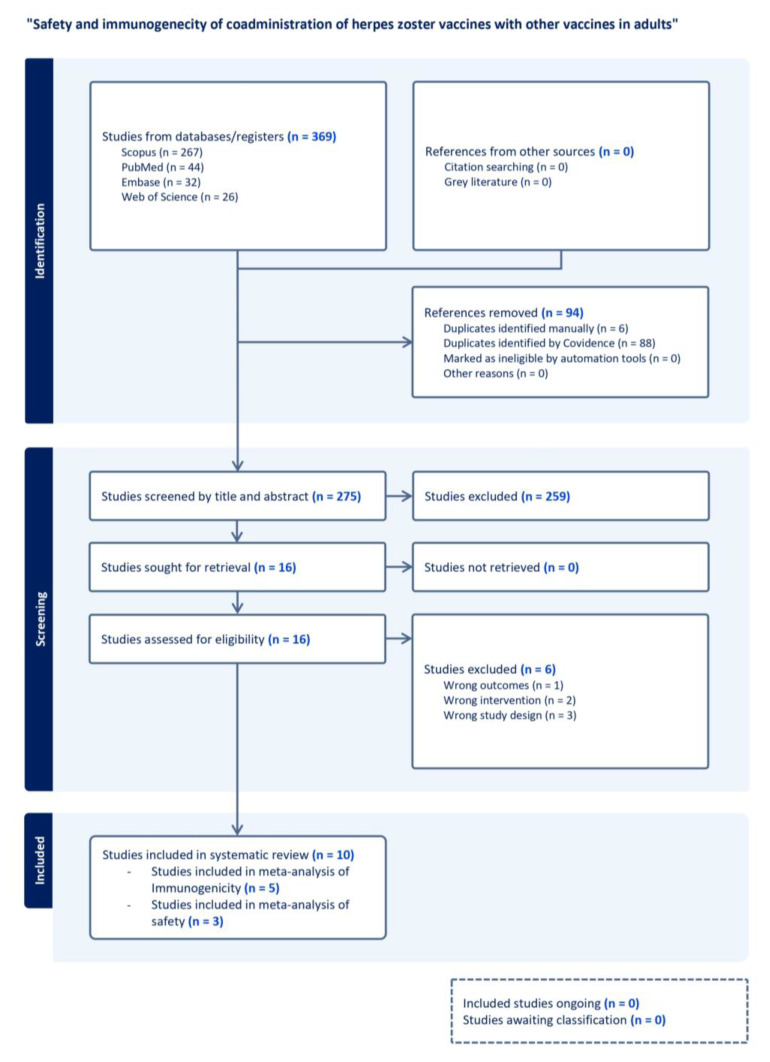
PRISMA flow dDiagram for study selection on the safety and immunogenicity of herpes zoster vaccine co-administration with other vaccines in adults. The diagram illustrates the systematic process for identifying and screening. It includes studies in the review on the safety and immunogenicity of co-administration of herpes zoster (HZ) vaccines with other vaccines in adults. A total of 369 studies were identified from database searches, with 94 duplicates removed. Of the 275 studies screened by title and abstract, 259 did not fulfill inclusion criteria and were excluded. Full texts of 16 studies were assessed for eligibility, with six excluded for wrong outcomes, interventions, or study design. Ten studies were included in the final systematic review.

**Figure 2 vaccines-13-00637-f002:**
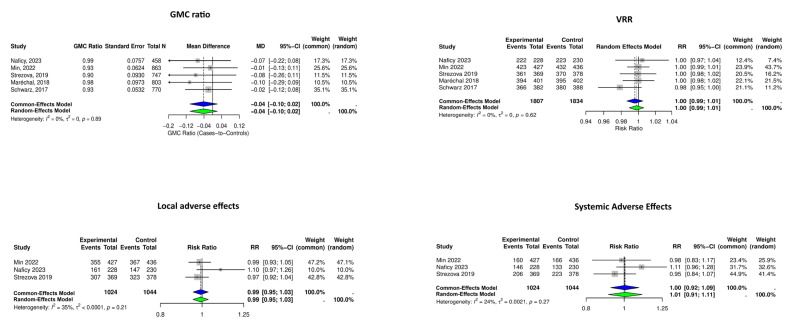
Forest plots summarizing the meta-analysis of immunogenicity and adverse effects of recombinant herpes zoster vaccine (RZV) following single- or co-administration. Geometric mean concentration (GMC) ratios, vaccine response rates (VRRs), and adverse effects did not increase when RZV was co-administered with other vaccines. Diamonds represent pooled estimates (blue: Common-Effects Model; green: Random-Effects Model) [10,12,13,16,18].

**Figure 3 vaccines-13-00637-f003:**
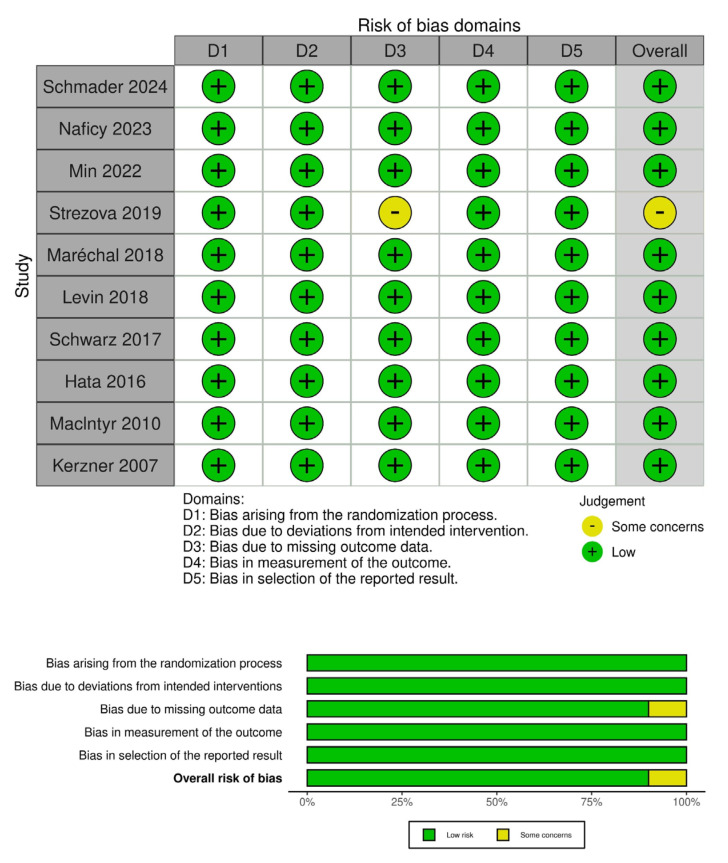
The risk of bias assessment for included studies using the RoB 2 tool for randomized studies [3,8,10,12,13,14,15,16,17,18]. The top panel displays the risk of bias across five domains for each study: D1 (randomization process), D2 (deviations from intended interventions), D3 (missing outcome data), D4 (measurement of the outcome), and D5 (selection of the reported result). The bottom panel summarizes the distribution of risk across all included studies, presenting the proportion of studies rated as “low risk” or “some concerns” in each domain and overall. Nine out of the ten studies were judged to have a low risk of bias across all domains, with just one showing some concerns related to missing outcome data.

**Table 1 vaccines-13-00637-t001:** Characteristics of the included studies.

First Author/Publication Year/Country	Study Design	Vaccines Used	Inclusion Criteria	Exclusion Criteria	Number of Participants
Schmader/2024/USA [10]	RCT	RZV, aIIV4, and HD-IIV4	Adults aged 65 years or older, community-dwelling, no immunosuppression, no dementia, able to speak English, no contraindications to the influenza or zoster vaccine	Participants were excluded if they recently received influenza, recombinant zoster, or COVID-19 vaccines, had acute or chronic conditions affecting immunity, or had contraindications to the study vaccines. Additional exclusions included recent hospitalization, febrile illness, certain medical histories (e.g., Guillain–Barré syndrome, active neoplastic disease), or conflicts of interest with study personnel.	267
Naficy/2023/USA [11]	RCT	RZV, mRNA-1273 COVID-19 booster vaccine	Adults aged = 50 years, completed 2-dose mRNA-1273 primary vaccination series at least 6 months prior to study vaccination, healthy or medically stable	Adults were excluded if they had medical conditions posing additional risks, a history of hypersensitivity to study vaccine components, or significant immune, cardiovascular, pulmonary, hepatic, or renal abnormalities. Other exclusions included recent investigational product use, immunosuppressive treatment, concurrent study participation, or prior herpes zoster or certain COVID-19 vaccinations, except as specified by the protocol.	539
Min/2022/USA [12]	RCT	RZV, PCV13	Adults aged = 50 years, provided written informed consent, able to comply with study requirements	History of herpes zoster, documented pneumococcal infection within the past five years, prior/planned administration of any HZ or pneumococcal vaccine other than study vaccine, cerebrospinal fluid leaks, cochlear implants, chronic renal failure, nephrotic syndrome, functional/anatomic asplenia.	912
Strezova/2019/USA [13]	RCT	RZV, Tdap	Adults aged = 50 years, provided written informed consent, able to comply with study requirements	History of HZ, previous HZ, VZV, or Tdap vaccination, chronic immunosuppressive therapy.	904
Maréchal/2018/Belgium [14]	RCT	RZV, PPSV23	Adults aged = 50 years	Previous vaccination against pneumococcal, VZV, or HZ, history of HZ, immunosuppressive therapy, certain medical conditions (e.g., cerebrospinal fluid leaks, cochlear implants, chronic renal failure, nephrotic syndrome, functional or anatomic asplenia).	865
Levin/2018/USA [15]	RCT	LZV, Influenza (IIV4)	Adults aged = 50 years with a history of varicella or residence in a VZV-endemic country for =30 years	Hypersensitivity to vaccine components, history of HZ, prior receipt of any varicella or zoster vaccine, receipt of an influenza vaccine for the 2015–2016 influenza season.	882
Schwarz/2017/USA [16]	RCT	RZV, Influenza (IIV4)	Adults aged = 50 years	Prior receipt of influenza vaccine, long-term treatment with immunosuppressant drugs or immune-modifying drugs within 6 months, prior VZV or HZ vaccination, history of HZ.	828
Hata/2015/Japan [3]	RCT	LVZ (live, attenuated Oka varicella vaccine^®^), PPSV23	Adults aged = 50 years with diabetes mellitus, no history of varicella or herpes zoster, seronegative for varicella-zoster virus antibodies	Previous vaccination against varicella or herpes zoster, immunocompromised state, pregnancy, hypersensitivity to vaccine components.	54
MacIntyre/2010/USA [11]	RCT	LVZ, PPSV23	Adults aged = 60 years	Immunocompromised state, previous vaccination against zoster or pneumococcal disease, allergy to vaccine components, pregnancy.	473
Kerzner/2007/USA [17]	RCT	LVZ, Influenza	Adults aged = 50 years, varicella history-positive, herpes zoster history-negative men and women (women had to be post-menopausal or have a negative urine pregnancy test)	Hypersensitivity reaction to any component of the study vaccines, prior receipt of any varicella or zoster vaccination, prior receipt of influenza vaccine for the 2005/06 influenza season, recent receipt of immune globulin or blood products, no live or inactivated vaccine during the study period, known immune dysfunction, concomitant use of antiviral therapy with activity against herpesviruses, participation in an investigational drug or vaccine study within 30 days before vaccination.	762

**aIIV4** (adjuvanted inactivated influenza vaccine); **COVID-19** (coronavirus disease of 2019); **HD-IIV4** (high-dose inactivated influenza vaccine); **HZ** (herpes zoster); **IIV4** (quadrivalent inactivated influenza vaccine); **LZV** (live-attenuated zoster vaccine); **mRNA-1273** (Moderna COVID-19 vaccine mRNA platform); **PCV13** (13-valent pneumococcal conjugate vaccine); **PPSV23** (23-valent pneumococcal polysaccharide vaccine); **RCT** (randomized controlled trial); **RZV** (recombinant zoster vaccine); **Tdap** (reduced-antigen-content diphtheria-tetanus-acellular pertussis vaccine); **VZV** (varicella-zoster virus).

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
