# Peer review of "Safety and Immunogenicity of Co-Administration of Herpes Zoster Vaccines with Other Vaccines in Adults: A Systematic Review and Meta-Analysis"

_vaccines, 2025, doi:10.3390/vaccines13060637_

Round 1
Reviewer 1 Report
Comments and Suggestions for Authors
I was invited to revise the paper entitled "Safety and immunogenicity of co-administration of herpes zoster vaccines with other vaccines in adults: A Systematic Review and Meta-analysis". It was a systematic review aimed to evaluate the efficacy and safety of VZ vaccines co-administred with other routine vaccinations. The topic is relevant for public health and the paper can improve the knowledge on this field. In particular, the evaluation of co-administration safety is crucial for the routine clinical practice.
Observation:
- Introduction section was poor and it needs improvements. In particular, Authors should better describe type of HZ vaccination available. Information on age indication and needs of co-administration should be introduced;
- In PRISM flow-diagram Authors should add the number of study included in the meta-analysis;
- Discussions should be improved. Some recent paper should be cited. In particular a recent meta-analysis on recombinant HZ vaccine immunogenicity should be discussed (10.3390/vaccines12050527;
- Authors should also discuss the potencial impact of co-administration of two adjuvanted vaccines on adverse events.
Author Response
Thank you for your time and the constructive comments on our manuscript.
Reviewer 1:
Comments and Suggestions for Authors
I was invited to revise the paper entitled "Safety and immunogenicity of co-administration of herpes zoster vaccines with other vaccines in adults: A Systematic Review and Meta-analysis". It was a systematic review aimed to evaluate the efficacy and safety of VZ vaccines co-administred with other routine vaccinations. The topic is relevant for public health and the paper can improve the knowledge on this field. In particular, the evaluation of co-administration safety is crucial for the routine clinical practice.
Reviewer 1, Comment 1:
Introduction section was poor and it needs improvements. In particular, Authors should better describe type of HZ vaccination available. Information on age indication and needs of co-administration should be introduced;
Response to Reviewer 1, Comment 1:
Thank you for highlighting this point. We have expanded the Introduction to provide a more detailed description of the available herpes zoster (HZ) vaccines.
On page 2, lines 60-72, it read:
“Therefore, preventing HZ is important, and international guidelines recommend HZ vaccination for immunocompromised adults aged >18 years and all adults aged ≥50 years (Anderson et al., n.d.). Currently, two HZ vaccine platforms have been approved: the live-attenuated VZV vaccine (LZV, Zo-tavax ®, MSD) and the adjuvanted VZV glycoprotein E (gE) subunit vaccine (rVZV, Shingrix, GSK) (Harbecke et al., 2021; Hata et al., 2016; Quan et al., 2024).
Individuals who receive HZ vaccines are also usually candidates to receive other vaccines, such as seasonal influenza, pneumococcal, and COVID-19 (Freedman et al., 2021).”
It now reads:
“Therefore, preventing HZ is important, and international guidelines recommend HZ vaccination for immunocompromised adults aged >18 years and all adults aged ≥50 years (Anderson et al., n.d.). Currently, two HZ vaccine platforms have been approved: the live-attenuated VZV vaccine (LZV, Zostavax ®, MSD) and the adjuvanted VZV glycoprotein E (gE) subunit vaccine (rVZV, Shingrix, GSK) (Harbecke et al., 2021; Hata et al., 2016; Quan et al., 2024). Zostavax ® is indicated for immunocompetent adults aged ≥50 years, but due to its live attenuated nature, it is contraindicated for immunocompromised individuals. In contrast, Shingrix, a non-live recombinant vaccine, is recommended for adults aged ≥50 years and immunocompromised adults aged ≥18 years due to its favorable safety profile and immunogenicity in these populations (Harbecke et al., 2021; Hata et al., 2016; Quan et al., 2024).
Due to the overlap in targeted age groups, immune status, and clinical recommendations, Iindividuals who receive HZ vaccines are also usually candidates to receive other vaccines, such as seasonal influenza, pneumococcal, and COVID-19 (Freedman et al., 2021). Co-administration not only simplifies vaccination schedules, and enhances patient compliance and convenience, but also improves vaccine uptake and coverage rates, which are essential for effective prevention of infectious diseases in vulnerable populations (Bonanni et al., 2023a). However, evidence regarding the immunological responses and adverse events following the co-administration of HZ vaccines with other vaccines in adults is limited.”
Reviewer 1, Comment 2:
In PRISM flow-diagram Authors should add the number of study included in the meta-analysis.
Response to Reviewer 1, Comment 2:
Thank you for this suggestion. We have updated the PRISMA flow diagram to explicitly indicate that out of the total studies reviewed, five were specifically included in the meta-analysis addressing immunogenicity, while three were included in the analysis evaluating adverse effects related to RZV, ensuring clarity and adherence to PRISMA reporting standards. Please see Figure 1.
Reviewer 1, Comment 3:
Discussions should be improved. Some recent paper should be cited. In particular a recent meta-analysis on recombinant HZ vaccine immunogenicity should be discussed (10.3390/vaccines12050527).
Response to Reviewer 1, Comment 3:
We appreciate this valuable suggestion. We have elaborated the discussion and referenced to the recent meta-analyses by Losa et al. and Omar Ali et al.
On page 11, lines 338-348, it read:
“We found and included RCTs that investigated the co-administration of RZV with influenza vaccines, COVID-19 mRNA vaccines, pneumococcal vaccines (PCV13, PPSV23), and Tdap. Our meta-analysis supports the grow-ing evidence that RZVs can be safely co-administered with other vaccines without compromising immunogenicity.”
It now reads:
“We found and included RCTs that investigated the co-administration of RZV with influenza vaccines, COVID-19 mRNA vaccines, pneumococcal vaccines (PCV13, PPSV23), and Tdap. Our meta-analysis supports the growing evidence that RZVs can be safely co-administered with other vaccines without compromising immunogenicity. Our findings are further supported by a recent comprehensive systematic review and meta-analysis by Losa et al., which specifically assessed the immunogenicity of RZV across various adult populations, including immunocompromised individuals (Losa et al., 2024). Notably, their analysis of within-study GMC comparisons found non-inferiority of humoral responses following RZV co-administration with other vaccines (COVID-19 mRNA-1273, PCV13, PPSV23, Tdap, and IIV4). Moreover, in line with our findings, Losa et al. observed no significant differences in VRRs related to co-administration (Losa et al., 2024). In another review, Omar Ali et al. included five RCTs and concluded that RZV co-administration does not significantly affect immunogenicity (Omar Ali et al., 2024). Although the aims, clinical questions, and methods used in the mentioned studies (Losa et al., 2024; Omar Ali et al., 2024) differed from ours, their findings further support that co-administered vaccines do not attenuate immune response to the RZV.
The absence of significant increases in adverse events in our study and the one by Omar Ali et al. (Omar Ali et al., 2024) is reassuring and supports using RZVs in multi-vaccine strategies. It is worth mentioning that the study by Schmader et al. was the only one that utilized RZV and adjuvanted influenza vaccine (aIIV4) and compared it to RZV and high-dose inactivated influenza vaccine (HD-IIV4) co-administration (Schmader et al., 2024). The RCT investigated safety but not immunogenicity and showed that these two vaccine combinations are safe and similar regarding the proportion of adverse effects (Schmader et al., 2024).”
Reviewer 1, Comment 4:
Authors should also discuss the potencial impact of co-administration of two adjuvanted vaccines on adverse events.
Response to Reviewer 1, Comment 4:
Thank you for this important consideration. We have added a dedicated paragraph discussing the potential implications of administering two adjuvanted vaccines simultaneously.
We elaborated on Page 11-12, line 352-374, which reads:
“The absence of significant increases in local and systemic adverse events in our study is reassuring and supports using RZVs in multi-vaccine strategies. RZV is an adjuvanted vaccine that incorporates the AS01 adjuvant system, which enhances innate immune activation and induces robust CD4⁺ T-cell and humoral responses (Roman et al., 2024). Other vaccines, such as the ad-juvanted influenza vaccine Fluad® Quadrivalent, adjuvanted hepatitis B vaccines (e.g., Heplisav-B®), and the RSV vaccine Arexvy®, also contain adjuvants. The co-administration of two adjuvanted vaccines may theoretically increase adverse events due to enhanced stimulation of the innate immune system. These effects may have clinical implications and could influence vaccine acceptance. Although such concerns are biologically plausible, current evidence to support this hypothesis is limited. A recent report from GSK presented preliminary results from the clinical trial NCT05966090, indicating acceptable reactogenicity following the co-administration of RZV and a RSV vaccine, though detailed data are not yet available.
The study by Schmader et al. was the only one that utilized RZV and adjuvanted influenza vaccine (aIIV4) and compared it to RZV and high-dose inactivated influenza vaccine (HD-IIV4) co-administration (Schmader et al., 2024). The RCT investigated safety but not immunogenicity and showed that these two vaccine combinations are safe and similar regarding the proportion of adverse effects (Schmader et al., 2024). Further studies are warranted to investigate the safety and immunologic interactions of co-administering two adjuvanted vaccines, especially in vulnerable populations.”

Reviewer 2 Report
Comments and Suggestions for Authors
Omid Rezahosseini et al have performed a meta-analysis of the safety and immunogenicity of co-administration of HZV with other vaccines. It is a complete set of data that supports the objective of the work. The reviewer would like to see this piece of work published in the Vaccine Journal.
It can be accepted after minor modifications.
Specific comments:
- Improve the quality of Figure 2.
- Figure 3 needs to be amended
- A detailed conclusion is needed
- Add a prospect in the revised manuscript
Author Response
Thank you for your time and the constructive comments on our manuscript.
Reviewer 2:
Omid Rezahosseini et al have performed a meta-analysis of the safety and immunogenicity of co-administration of HZV with other vaccines. It is a complete set of data that supports the objective of the work. The reviewer would like to see this piece of work published in the Vaccine Journal.
It can be accepted after minor modifications.
Specific comments:
Reviewer 2, Comment 1: Improve the quality of Figure 2.
Response to Reviewer 2, Comment 1:
Thank you for pointing this out. Figure 2 has been attached in higher resolution.
Reviewer 2, Comment 2: Figure 3 needs to be amended
Response to Reviewer 2, Comment 2:
We have thoroughly reviewed and amended Figure 3 for improved clarity and accuracy.
On page 10, it read:
“Figure 3. The risk of bias assessment for included studies using the RoB 2 tool for randomised studies. The first plot shows the risk of bias across five domains for each study. The second plot summarizes the proportion of studies at low risk or with some concerns across each domain and overall. Most studies demonstrated a low risk of bias, with some concerns noted in the domain related to missing outcome data.”
It now reads:
“Figure 3. The risk of bias assessment for included studies using the RoB 2 tool for randomised studies. The top panel displays the risk of bias across five domains for each study: D1 (randomization process), D2 (deviations from intended interventions), D3 (missing outcome data), D4 (measurement of the outcome), and D5 (selection of the reported result). The bottom panel summarizes the distribution of risk across all included studies, presenting the proportion of studies rated as “low risk” or “some concerns” in each domain and overall. Nine out of the ten studies were judged to have a low risk of bias across all domains, with just one showing some concerns related to missing outcome data.”
Reviewer 2, Comment 3: A detailed conclusion is needed
Response to Reviewer 2, Comment 3:
In line with your recommendation, we have expanded the conclusion to provide a comprehensive summary of our findings.
On page 12, it read:
“Our findings support the co-administration of RZVs with influenza vaccines, COVID-19 mRNA vaccines, pneumococcal vaccines (PCV13, PPSV23), and Tdap, showing no significant impact on immunogenicity or adverse events. LZVs, while generally safe, require further investigations. These findings align with prior research and provide actionable insights for vaccination strategies and policies to improve vaccine uptake and protection in adult populations.”
Now reads:
“Our findings support the co-administration of RZVs with influenza vaccines, COVID-19 mRNA vaccines, pneumococcal vaccines (PCV13, PPSV23), and Tdap, showing no significant impact on immunogenicity or adverse events in healthy adults. While co-administration of LZVs were generally safe, the limited number of studies, and exclusion of immunocompromised populations highlight the need for further investigation to confirm safety and immunogenicity in the vulnerable populations. Our findings are consistent with previous research and offer practical implications for public health policies, supporting flexible and efficient vaccination schedules that can improve coverage, reduce missed opportunities for immunization, and enhance protection in aging and vulnerable populations.”
Reviewer 2, Comment 4: Add a prospect in the revised manuscript
Response to Reviewer 2, Comment 4:
We appreciate your suggestion. On page 12, we have added a "Future Perspectives" section in the discussion.

Round 2
Reviewer 2 Report
Comments and Suggestions for Authors
Thanks for the corrections, now it can be accepted